# Mediterranean Diet Nutrients to Turn the Tide against Insulin Resistance and Related Diseases

**DOI:** 10.3390/nu12041066

**Published:** 2020-04-12

**Authors:** Maria Mirabelli, Eusebio Chiefari, Biagio Arcidiacono, Domenica Maria Corigliano, Francesco Saverio Brunetti, Valentina Maggisano, Diego Russo, Daniela Patrizia Foti, Antonio Brunetti

**Affiliations:** Department of Health Sciences, University “Magna Græcia” of Catanzaro, 88100 Catanzaro, Italy; maria.mirabelli@unicz.it (M.M.); echiefari@gmail.com (E.C.); arcidiacono@unicz.it (B.A.); domenicacorigliano@gmail.com (D.M.C.); francescosaverio.brunetti@studenti.unicz.it (F.S.B.); vmaggisano@unicz.it (V.M.); d.russo@unicz.it (D.R.); foti@unicz.it (D.P.F.)

**Keywords:** Mediterranean diet, insulin resistance, nutraceuticals, extra-virgin olive oil, HMGA1

## Abstract

Insulin resistance (IR), defined as an attenuated biological response to circulating insulin, is a fundamental defect in obesity and type 2 diabetes (T2D), and is also linked to a wide spectrum of pathological conditions, such as non-alcoholic fatty liver disease (NAFLD), cognitive impairment, endothelial dysfunction, chronic kidney disease (CKD), polycystic ovary syndrome (PCOS), and some endocrine tumors, including breast cancer. In obesity, the unbalanced production of pro- and anti-inflammatory adipocytokines can lead to the development of IR and its related metabolic complications, which are potentially reversible through weight-loss programs. The Mediterranean diet (MedDiet), characterized by high consumption of extra-virgin olive oil (EVOO), nuts, red wine, vegetables and other polyphenol-rich elements, has proved to be associated with greater improvement of IR in obese individuals, when compared to other nutritional interventions. Also, recent studies in either experimental animal models or in humans, have shown encouraging results for insulin-sensitizing nutritional supplements derived from MedDiet food sources in the modulation of pathognomonic traits of certain IR-related conditions, including polyunsaturated fatty acids from olive oil and seeds, anthocyanins from purple vegetables and fruits, resveratrol from grapes, and the EVOO-derived, oleacein. Although the pharmacological properties and clinical uses of these functional nutrients are still under investigation, the molecular mechanism(s) underlying the metabolic benefits appear to be compound-specific and, in some cases, point to a role in gene expression through an involvement of the nuclear high-mobility group A1 (HMGA1) protein.

## 1. Introduction

Insulin resistance (IR) has been defined as a dysmetabolic condition in which the peptide hormone insulin produces a less-than-expected biological effect on peripheral target tissues, leading to hyperinsulinemia, the diagnostic hallmark of IR [1]. IR affects approximately 25%–35% of Westernized populations [2] and is consistently associated with obesity [3] and obesity-related complications, such as type 2 diabetes (T2D) mellitus [4], cardiovascular disease (CVD) [5,6], certain types of cancer [7], infertility [8], non-alcoholic fatty liver disease (NAFLD) [9,10], and cognitive impairment [11]. Although surgical and pharmacological strategies [12] have been shown to be useful, energy reduced diets, as part of a healthy lifestyle, may almost invariably facilitate weight loss and reduce IR in these patients [13]. Even a moderate weight loss of only 5%–10% can lead to several health advantages in obese individuals, which include improvement in cardiometabolic parameters, reduction of blood pressure, and increase in longevity [13], in addition to positive changes in insulin sensitivity and inflammatory biomarkers [14,15,16]. As part of a hypocaloric regimen, the Mediterranean diet (MedDiet) has proved to be associated with a greater improvement of IR in obese individuals when compared to other low-energy dietary approaches [17], even after a mild decrement of body weight of less than 5% [18]. Also, the reduction in insulin levels and other measures of IR, i.e., the homeostatic model assessment (HOMA) index, triggered by this dietary approach, are precocious and sustainable over time [17,18]. These benefits should be ascribed not only to the caloric restriction state, which promotes weight loss and reduction of IR independent of the diet composition, but also to the large amount of functional foods and nutraceuticals naturally present within the MedDiet. The “Mediterranean diet” term reflects food combinations typical of Mediterranean populations, such as Greeks, Southern Italians and Spanish, in the early 1960s, in which adult life expectancy was found, together with the Japanese population, to be among the highest in the world, with lower rates of CVD and other chronic aging-related diseases [19]. Being predominantly plant-based, the MedDiet is characterized by a daily abundance of vegetables, legumes, whole grain bread and other cereals, nuts and seeds, fresh fruit as the typical dessert, extra virgin olive oil (EVOO) as the principal source of fat, a low to moderate consumption of dairy products, fish, poultry, and eggs, a low consumption of red meat, and a moderate consumption of wine, normally with meals [20]. As a consequence, this eating pattern is low in saturated fat (~10% of energy) [21], and rich in several minor functional components, including vitamins, carotenoids, unsaturated fatty acids, and various bioactive plant-derived phenolic compounds, depicted by antioxidant and anti-inflammatory properties, currently in the center of research interest [22]. Phenolic compounds from plant sources may also modulate insulin action and metabolism in insulin-sensitive tissues, with potential preventive or curative effects against IR and IR-related diseases. As recently evidenced in cross-sectional investigations, adherence to the Mediterranean eating style can give health advantages starting from young age [23]. The consumption of fish, nuts and dairy products, representing a fundamental aspect in the MedDiet, can be related to a better body composition and fitness performance in adolescents, in terms of speed/agility and cardiorespiratory endurance [23]. Nonetheless, even in European countries surrounding the Mediterranean Sea, half of the children and adolescents show only a low adherence to this healthy dietary tradition [24], with a high rate of pediatric overweight and development of obesity and related complications in adult life. Rising obesity rates will contribute to large healthcare expenditure increases, and this would require future cost containment efforts through large-scale health promotion programs and weight loss interventions, fostering a combination of diet quality changes and physical activity [25]. In the last decade, research into the biological properties of the Mediterranean functional components has led to the isolation of several natural active ingredients, followed by the production of distinct semi-synthetic nutritional supplements as alternatives for pharmaceuticals for health purposes. Recent in vivo studies, in either experimental animal models or clinical settings, have shown encouraging results for distinct insulin-sensitizing nutraceuticals derived from Mediterranean food sources in many IR complications. In the present narrative review, much effort has been devoted to present mechanistic and epidemiological evidences about the role of the MedDiet and selected nutritional supplements on hard curative IR-related diseases consistently ranked among major CVD risk factors, such as T2D, NAFLD, endothelial dysfunction, dyslipidemia and chronic kidney disease (CKD), along with its beneficial effects on cognitive impairment, polycystic ovary syndrome (PCOS) and breast cancer.

## 2. Mediterranean Diet (MedDiet) and Unsaturated Fatty Acids for Managing Type 2 Diabetes (T2D)

T2D, a frequent consequence of IR, is the most common metabolic disease, affecting millions of people throughout the world [26]. In T2D, chronic hyperglycemia is associated with a variety of co-morbidities, such as dyslipidemia, hypertension, micro- and macrovascular complications, which profoundly impair the quality and expectancy of life for affected patients, with huge costs for the health system [27]. Dysfunctional adipose tissue in obesity is associated with hypertrophy and hyperplasia of adipocytes, chronic inflammatory cell infiltration, and activation of the cytokine network, representing the main determinant of IR and the major risk factor for the development of T2D and its complications [28]. At first, in obese individuals destined to become diabetics, pancreatic β-cells may compensate for the impaired peripheral insulin responsiveness by secreting more insulin into the bloodstream in an effort to reduce blood glucose levels, thus leading to hyperinsulinemia, a biological marker of IR, often indicative of a pre-diabetic status [29,30]. Then, in these individuals, the full onset of T2D is elicited by the gradual loss of pancreatic β-cell mass and dysfunction which occur over time and prevent further hyperinsulinemic compensation. Fortunately, the progressive nature of T2D can be challenged by precocious interventions, based on intensive lifestyle changes and/or daily metformin therapy, which can successfully prevent or delay the onset of diabetes among high-risk individuals by over 50% [31]. However, the role of diet and nutrition is crucial in the overall management of T2D, which is far from being limited to prevention. As stated in the American Diabetes Association (ADA) guidelines [32], there is not a one-size-fits-all nutritional approach for individuals affected by T2D, and meal planning, delivered by registered dieticians, should be individualized, taking into account current eating patterns, preferences, and specific glycometabolic goals. With this in mind, the MedDiet [33,34], and other plant-based [35,36] nutritional approaches, all associated with positive outcomes in clinical research conditions, can be endorsed as suitable treatment measures to achieve glycemic control, so that the need for pharmacological medications would be minimized. In particular, important health benefits have been demonstrated for the Mediterranean eating pattern, which can considerably improve fasting glucose, glycated hemoglobin (HbA1c) and insulin levels in obese diabetic patients, when compared to low-fat diets [17]. These glyco-metabolic benefits appear to be sustainable over time, independent of body weight change, as they do not occur with other successful nutritional interventions for weight loss [17]. Furthermore, as evidenced in several randomized trials, adherence to typical dietary habits of the Mediterranean countries, may help in reducing all-cause and CVD-related mortality in patients with T2D [37,38], which is consistent with the proof of anti-hypertensive effects for specific nutrients of the MedDiet [39]. Monounsaturated fatty acids (MUFAs) and polyunsaturated fatty acids (PUFAs), in forms of olive oil, nuts and seeds in plant-based dietary regimens, may function as the driving force for the amelioration of glucose metabolism, insulin sensitivity, blood lipids and CVD risk, observed either in diabetic individuals [17,40,41] or in the general population [42]. Confirmation of the antidiabetic effect of these nutrients has been given by a meta-analysis of 24 randomized controlled trials, evidencing better glycemic control, serum lipid profile, and systolic blood pressure among diabetics on high-MUFAs or high-PUFAs plant-based diets over low-fat, high-carbohydrate ones [43]. Moreover, the abundant intake of MUFAs or PUFAs with olive oil and seeds, in place of saturated or trans fatty acids, may also reduce the risk of T2D in at-risk patients by up to 83% over a median of 4.4 years [44], even in the absence of caloric-restriction [45]. As shown in animal model studies [46,47,48], PUFAs may ameliorate the adipose tissue’s inflammatory responses, with beneficial effects on insulin sensitivity [49]. Given that pro-inflammatory cytokines and chemokines, like tumor necrosis factor alfa (TNF-α), interleukin-6 (IL-6) and resistin, overproduced by the dysfunctional adipose tissue in obesity, can activate intracellular pathways that trigger IR in insulin-target tissues [15], the anti-inflammatory potential of PUFAs may indirectly improve the peripheral insulin responsiveness, reducing the risk of glyco-metabolic alterations in patients with IR [49]. Additionally, PUFAs can bind and stimulate G-protein-coupled receptors (GPCRs) which play important roles in regulating glucose metabolism, such as GPR120, thereby leading to increased secretion of the glucagon-like peptide 1 (GLP-1) hormone from enteroendocrine L-cells [50]. By stimulating insulin release from pancreatic β-cells, with immediate consequences of increased glucose uptake from skeletal muscles, raised GLP-1 levels may in turn limit postprandial hyperglycemia [51]. Also, GLP-1 may influence satiety at central nervous system level, attenuating appetite sensations and, thus, the amount of food consumed with meals, so that energy intake does not exceed expenditure. Figure 1 provides a schematic representation of how PUFAs can influence glycemic control.

From a diabetes prevention perspective, the GLP-1-releasing effects of PUFAs could be particularly important, given that obese patients have lower GLP-1 responses to oral glucose than individuals of normal weight, irrespective of the glycemic status [52,53]. Nonetheless, a large meta-analysis of controlled trial, has recently evidenced that dietary supplementation with PUFAs may have little or no effect on TD2 prevention, or measures of IR, in at-risk patients [54]. However, background eating patterns and total fat intake, which were not reported, could have affected the results. In fact, favorable effects of substituting unsaturated fatty acids for saturated fatty acids on insulin sensitivity can be seen only at a total fat intake below 37% of energy [55], whereas higher quantity of fat intake increases the risk of IR independently of quality. Finally, the positive properties of the MedDiet for prevention and management of T2D are not confined within selected PUFA- and/or MUFA-enriched functional foods, but incorporate additional and synergistic benefits from the ingestion of numerous promising bioactive polyphenol compounds [56], which are currently under investigation.

## 3. MedDiet Flavonoids for Preventing T2D

Dietary flavonoids represent a large and heterogeneous group of polyphenols ubiquitously found in daily consumed fruits and wine, as well as vegetables, nuts, cocoa, tea and grain seeds [57]. On the basis of their chemical structure, flavonoids can be categorized as flavonols, flavones, flavan-3-ols, anthocyanins, flavanones, and isoflavones [57]. Recently, a meta-analysis of 8 prospective cohort studies, including a total of 312,015 participants, of whom 19,953 developed T2D during 4–28 years of follow-up, investigated the role of regular consumption of flavonoids in the prevention of diabetes, taking into account age, sex, total energy intake, body mass index (BMI), smoking, alcohol intake, and physical activity [58]. Compared with the lowest exposed group (8.9 to 501.8 mg/day), the group with the highest intake of total flavonoids (33.2 to 1452.3 mg/day) was associated with a 11% decreased risk of developing T2D during follow-up. Additionally, the dose-response analysis suggested a 5% reduction of T2D risk for each 300 mg/day increment of total flavonoids intake [59]. With regard to flavonoid subclasses, a beneficial effect in reducing the risk of T2D was significant for anthocyanidins, flavan-3-ols, flavonols, and isoflavones, but not for other subclasses, suggesting that different bioactive molecular pathways may be triggered by these nutrient compounds in relation to their chemical structure. By way of illustration of their speculated pharmacological antidiabetic mechanisms, we propose the lead berries-extracted quercetin, which acts as the base for the formation of other flavonol skeletons [57]. Similar to resveratrol, other flavonoids and the biguanide metformin that is recommended as the first-line intervention for diabetes treatment and prevention, quercetin can activate the insulin-independent 5′ adenosine monophosphate-activated protein kinase (AMPK) pathway of skeletal muscle cells, slowing the oxygen consumption of adenosine diphosphate in isolated mitochondria [59]. Also, quercetin can enhance the uptake of glucose in skeletal myocytes through an AMPK-dependent up-regulation of glucose transporter GLUT-4, which may occur independently of the insulin signaling under oxidative stress conditions [60]. In addition to this, several in vitro studies have confirmed that flavonoids may possess strong inhibitory activities on the intestinal α-glucosidase enzyme that catalyzes the cleavage of glucose from disaccharides, thereby delaying the absorption of glucose and flattening postprandial hyperinsulinemic/hyperglycemic excursions [61,62].

## 4. MedDiet and Extra-Virgin Olive Oil (EVOO)-Derived Secoiridoids for Treating Non-Alcoholic Fatty Liver Disease (NAFLD)

NAFLD is an excessive fat deposition in the liver in the absence of secondary causes, which commonly occurs in people with obesity and IR. Estimates of the prevalence of NAFLD range from 25% in the general population [63] to over 50% in people with T2D [64]. However, the term “NAFLD” encompasses a wide spectrum of liver diseases, ranging from fat infiltration of the liver with minimal inflammation, known as steatosis, to nonalcoholic steatohepatitis (NASH), which consists of a lobular inflammatory cell infiltrate with hepatocyte ballooning in the absence or presence of fibrosis, to end-stage liver disease or cirrhosis. Approximately 25%–30% of individuals with NAFLD can progress to NASH [65], whereas the same proportion of individuals with NASH can progress to cirrhosis and hepatocellular carcinoma, especially those with liver fibrosis [66,67]. However, beyond liver-related morbidity and mortality, patients with NASH have also an increased risk of CVD and CVD-related death [68]. Given its high prevalence in the general population and the potential serious risks on health outcomes, NAFLD should be treated immediately upon diagnosis, encouraging dietary modifications as the first-line strategy to achieve the loss of 5%–10% of the initial body weight. In patients with NAFLD, a modest weight loss can produce a significant decrease in liver fat content, with improved serum aminotransferase activity and beneficial health effects [69,70]. Nonetheless, it is worth mentioning that different weight loss goals can be set for the management of NAFLD, depending on the detection of NASH in individual patients. In fact, while for patients with steatosis, loss of 5%–7% of the initial body weight can suffice, in those with suspected or biopsy-proven NASH higher weight loss goals (7%–10%) should be recommended [71]. Also, careful attention should be paid to avoid fast and excessive weight loss in patients with NAFLD. In this regard, very low-calorie diets, providing about 800 or less kcal per day, are not generally advisable, due to the risk of protein-calorie malnutrition and potential exacerbation of liver fibrosis and necrosis [72,73]. Although several dietary approaches could be used for the treatment of NAFLD [74], over the past several years, the Mediterranean one has attracted special interest. Cross-sectional and longitudinal reports evidenced a lower likelihood of NASH in patients who were adherent to the MedDiet [75,76], whereas randomized controlled crossover trials defined the superiority of this diet in the improvement of insulin sensitivity, metabolic parameters and steatosis over low-fat ones [77,78]. Furthermore, a 6-month nutritional counseling to adhere to the MedDiet has proved to be effective in the amelioration of certain disease-specific traits, including liver imaging, liver fibrosis score, inflammatory/oxidative biomarkers and glycemic status indices in non-fibrotic NAFLD patients [79]. Apart from negative variations in circulating visfatin levels, the longitudinal phenotype changes were more pronounced among individuals carrying the signal transducer and activator of transcription 3 (STAT3) rs2293152 “G” allele, which has been linked with greater NAFLD susceptibility and severity [79]. As such, the MedDiet eating pattern, which is rich in unsaturated fatty acids and plant-based polyphenols, has recently emerged as the most appropriate nutritional approach for NAFLD, gaining the support of the European Association for the Study of Liver (EASL) and the European Association for the Study of Diabetes (EASD) [80]. EVOO, which is the major source of fat in the MedDiet, is probably, the main driver of its beneficial effects on liver function and structure. Several studies, in this context, have investigated the biological activities of polyphenolic compounds in EVOO, such as oleuropein and its derivative secoiridoids, either in vitro or in vivo, in experimental animal models [81,82,83,84]. Recently, it has been given importance to oleacein, an abundant lipophilic degradation product of oleuropein, present in EVOO at higher concentration and provided with an increased bioavailability, due to higher resistance to the acidic gastric environment and better intestinal absorption [85]. Additionally, the semi-synthesis of oleacein has proven to be sustainable [86], laying the foundation for the future implementation of novel functional foods and nutraceuticals. Although this concept is still at early developmental stages, mice on high fat diet (HFD), treated with 20 mg/day oleacein for 5 weeks, were protected from abdominal fat accumulation, weight gain, and liver steatosis when compared to untreated controls, with evidence of improved insulin action on the liver and preserved glucose and lipid homeostasis [87]. Some of the molecular constituents, that appear to be involved in oleacein action, include some known nutrient-responsive regulators of lipid metabolism: the transcriptional activator sterol regulatory element-binding transcription factor-1 (SREBP-1), and its target fatty acid synthase (FAS), whose protein levels were significantly reduced following oleacein treatment in liver and fat [85,87]. Also, phospho-ERK, a serine/threonine kinase downstream effector of the mitogen-activated protein kinase (MAPK) cascade, was inhibited in liver tissue of treated mice, thereby suggesting greater responsiveness to insulin and less severity of IR, in situations of chronic high-fat hypernutrition [88].

## 5. MedDiet and Purple Plant-Derived Anthocyanin Extracts for Neuroprotection

Increasing evidence is linking adiposity to impaired brain structure, with proof of frontal gray matter atrophy across all ages in patients with excessive body weight [89,90]. Also, obese individuals suffering from T2D, have higher levels of cerebrovascular disease, smaller total and regional brain volumes, decreased cerebral connectivity and metabolic brain networks [91], increased β-amyloid deposition and tau phosphorylation, accelerated rates of cognitive impairment and, especially in women, higher risk of developing Alzheimer’s dementia [92]. This latter risk is directly proportional to the duration and magnitude of hyperglycemia in T2D, but is also detectable in pre-clinical stages of T2D, during compensatory hyperinsulinemia and peripheral IR. In this context, it has been proposed that in patients with declining cognitive abilities, from premorbid levels to clinical evidence of dementia, with or without comorbid T2D, the brain itself becomes insulin resistant. As such, like insulin-sensitive peripheral tissues, the insulin signaling and cerebral structure may be influenced by dietary energy content and food composition [93]. Hypocaloric diets and/or diets of higher quality, with low fat and sugar intakes, are associated with larger brain volumes, better white matter integrity, and better cerebral health [94] in animal models. Although the human evidence is less consistent, adherence to the MedDiet may reduce the progression of well-established neuroimaging biomarkers of cognitive impairment, including β-amyloid deposition and cerebral glucose utilization via positron emission tomography, and provide 1.5 to 3.5 years of protection against Alzheimer’s dementia, in middle-aged cognitively normal individuals [95]. Also, cross-sectional studies have found positive associations between higher adherence to the MedDiet and larger brain volumes [96,97,98], cerebral connectivity [99], or fewer white matter lesions [100] at older ages, whereas hypercaloric diets, high in meat, carbohydrate and sugars intake and low in fish and vegetables contents, are associated with brain atrophy [94] and cortical thickness [101]. Discrepancies in long-term neurological outcomes between these two nutritional styles could be, at least in part, explainable by the different amounts of flavonoids, a class of phenolic compounds widely distributed in plants. In particular, one of the major flavonoid subclasses, known as anthocyanins, which are responsible of the red, blue, and purple pigmentation of many vegetables and fruits, have earned significant attention in the context of neurodegenerative diseases [102]. After entering the brain tissue, anthocyanins tend to accumulate [103,104,105], with the potential of high local bioavailability for neuroprotective actions [105]. The efficacy of anthocyanins has been assessed either in vitro or in vivo, using various animal models of early-onset Alzheimer’s disease, such as the APP/PS1 double transgenic mouse [106,107], which overexpresses both a mutant human amyloid precursor protein (APP) and a mutant human presenilin 1 (PS1) and is prone to cognitive impairment due to increased cerebral deposition of β-amyloid. Treatments with either total anthocyanins extracted from bilberry and black currant [106] or isolated cyanidin-3-O-glucopyranoside, which is the predominant anthocyanin in colored fruit and vegetables [107], were able to ameliorate cognitive functions in APP/PS1 mice. As summarized in Figure 2, the beneficial cognitive effects of anthocyanins are related, at least in part, to the improvement in processing of beta amyloid and neuroinflammation, which follows the activation of peroxisome proliferator-activated receptor γ (PPARγ) [106].

PPARγ belongs to a nuclear hormone receptor superfamily of ligand-inducible transcription factors that heterodimerize with the retinoid X receptor (RXR) [108] and bind to peroxisome proliferator response elements (PPREs) in the promoter region of target genes [109]. Although present in most cell types, such as vessels, neurons, macrophages, microglia and astrocytes, in which it attenuates the expression of proinflammatory mediators [110], PPARγ is predominantly expressed in adipocytes, wherein modulates lipid metabolism in form of release, transport, and storage of free fatty acids (FFAs) [111,112]. By enhancing the uptake and storage of FFAs in adipose tissue, PPARγ agonists may decrease ectopic fat accumulation [113], thus preserving non-adipose peripheral tissues from the wide range of lipotoxicity complications, including IR, liver steatosis, hyperglycemia and CVD [114]. However, peripheral IR is strongly related to brain dysfunction, either due to reduced insulin transport into the brain [115] or to altered local insulin receptor sensitivity and activation [116]. Notwithstanding a marginal role in neuronal glucose uptake under basal conditions, as it mostly occurs in an insulin-independent manner, insulin positively regulates the normal brain function, particularly by enhancing the spatial working memory via the insulin-sensitive glucose transporter GLUT-4 [117]. In line with these considerations, cyanidin-3-O-glucopyranoside, by acting as a potent natural agonist for PPARγ with peripheral and central insulin-sensitizing effects, has been shown not only to reduce liver fat [118] and fasting glucose concentrations [107] of treated animals, but also to increase cerebral glucose uptake [107]. Furthermore, promising clinical results about the impact of anthocyanin consumption in elderly individuals with mild cognitive impairment has emerged recently. In these patients, the dietary introduction of a daily anthocyanin-rich fruit juice significantly improved verbal fluency, as well as short-term and long-term memory, over a 12-week period [119], which is consistent with other reports [120,121]. Nonetheless, it cannot be excluded that some benefits on cognitive processes may have been caused by detrimental effects of nutritional deficiencies in control subjects, rather than the surplus consumption of anthocyanins in interventional groups. However, the incorporation of functional beverages into normal dietary patterns might still be a practical and convenient strategy to increase the daily intake of phytochemicals with insulin sensitivity-promoting effects in the general population, closing the gap between the actual vegetable intake and clinical recommendations [122]. On a different note, the beneficial cognitive effects of the MedDiet in elderly patients with dementia may be augmented by the isocaloric dietary supplementation of medium-chain saturated fatty acids in the form of coconut oil, which correlates with an immediate increase of circulating ketone bodies [123]. Brain exposure to ketone bodies can replace the brain’s normal reliance on glucose and potentially reverse the pathological alterations of Alzheimer’s disease, ultimately improving the cognitive performances in affected patients, with a greater impact on women than men, after only 21 days of coconut oil supplementation [123].

## 6. MedDiet and Resveratrol in Polycystic Ovary Syndrome (PCOS)

PCOS, characterized by hyperandrogenism, chronic anovulation and/or sonographic evidence of small cysts in one or both ovaries, is the most common endocrinological disorder among women of reproductive age and the main cause of female infertility [124]. Although the etiology is still unknown, obesity, IR and hyperinsulinemia are cardinal features of PCOS [124]. Insulin regulates the ovarian function through interactions with gonadotropins, and, in case of raised levels, amplifies the luteinizing hormone (LH)-induced ovarian androgen production, ultimately preventing ovulation [125]. Besides a well-established role of weight gain in the pathophysiology of IR, there is evidence that ovarian dysfunction in PCOS may be further enhanced by specific nutritional deficiencies and high dietary contents of sugar and fat [126,127]. Also, it should be noted that obese women with PCOS-related infertility have also been shown to engage in inappropriate eating behaviors, which can often interfere with traditional weight-loss programs [128,129]. On the other hand, in PCOS women, even a modest weight reduction may exert significant beneficial effects on IR, hyperandrogenism and menstrual problems, allowing the complete resolution of all symptoms of PCOS in some cases [130]. The adherence to hypocaloric low-carbohydrate, low-fat diets may induce both short and long-term reductions in fasting and post-challenge insulin concentrations, as markers of increased insulin sensitivity, that can over time improve menstrual regularity and reproductive outcomes in obese women with PCOS [131,132]. Preliminary evidence of positive metabolic and endocrinological outcomes, in terms of reduced body weight, free testosterone, LH and fasting insulin levels, has been also reported on a calorie-restricted, very-low carbohydrate, ketogenic diet (VLKD) [133]. On the other hand, it is still uncertain for how long a very-low carbohydrate dietary pattern can be followed to achieve the best outcome without resulting in potential safety concerns [133,134]. In contrast, the effects of the MedDiet on PCOS outcomes have attracted intense interest in the last few years [135,136]. Based on the current view, the dietary management of PCOS, which relies on energy restriction and a Mediterranean nutritional approach, is deemed to have a beneficial impact on some reproductive and metabolic parameters, including menstrual regularity, blood pressure, glucose homeostasis, lipid profile and estimates of CVD risk [137]. In this context, several dietary biomolecules, widely found in Mediterranean foods, are postulated to be responsible for the amelioration of distinct PCOS traits. For example, the polyphenol stilbene derivative resveratrol, found in grape seed, red wine and some berries, may appease the hyperandrogenic traits of PCOS (Figure 3).

In a randomized placebo-controlled trial, treatment with 1500 mg daily of resveratrol significantly decreased testosterone and dehydroepiandrosterone sulfate excess by ~23% in PCOS women over a period of 3 months [138]. Changes in androgen levels could have been explained by the direct inhibitory effects of resveratrol on ovarian theca-interstitial proliferation and expression of 17α-hydroxylase/C17-20-lyase, known as the rate-limiting enzyme in androgen biosynthesis, which were demonstrated in in vitro experimental models [139,140]. However, significant decrement in insulin concentrations and a consistent increase of insulin sensitivity, in terms of Matsuda insulin sensitivity index values, were noted in PCOS women after treatment with resveratrol [138]. Given that chronic hyperinsulinemia can stimulate an excessive production of androgens in PCOS women, the antiandrogenic effects of resveratrol are probably indirect and driven by the reduction of IR in these patients [141,142,143,144], in contrast to pharmacologic intervention with metformin [145,146]. Disregarding this special effect, the insulin-sensitizing actions of resveratrol are complex and not completely elucidated. Part of the pharmacodynamic effects of this stilbene compound could derive from its capacity to modulate different pathways and molecular targets, including those downstream of the insulin receptor and mediated by AMP-activated protein kinase (AMPK) and sirtuin 1, as reported in clinical studies on diabetic patients and rodent models of IR, as well as in vitro experimental settings [147,148,149,150,151]. In these studies, treatment with resveratrol has proved to activate both fuel-sensors AMPK and sirtuin 1 in insulin-sensitive peripheral tissues, especially skeletal muscles. Sirtuin 1 is a nicotinamide adenine dinucleotide (NAD^+^)-dependent histone deacetylase involved in the regulation of mitochondrial biogenesis, inflammation, intracellular metabolism, apoptosis and glucose homeostasis, whereas AMPK is a serine/threonine protein kinase complex, that, following activation by caloric starvation and ATP depletion, regulates fatty-acid oxidation, glucose uptake and mitochondrial function and biogenesis, in partnership with sirtuin 1. Based on these considerations, nutritional interventions with AMPK activators, such as resveratrol and other plant-based polyphenols, may mimic caloric restriction and secure similar effects on health and wellbeing as a calorie-deprived dietary regimen [152], potentially offsetting safety limitations and sustainability of long-term very-low-energy diet programs for weight loss, typically encountered in clinical practice.

## 7. MedDiet and Nutrient–Gene Interactions in the Modulation of Breast Cancer Risk

A large body of epidemiological evidence indicates an association between T2D and increased risk of developing some common female malignancies, including breast, endometrial and ovarian cancers [7,153]. Also, even in non-diabetic populations, the presence of obesity-related IR can increase the risk of malignancies in both premenopausal and postmenopausal phases of a woman’s life [154]. Breast cancer is the most commonly occurring cancer (over 2 million incident cases in 2018) in women, and the second most diagnosed cancer overall, following lung cancer [155,156]. Breast malignancies are also the leading cause of worldwide cancer mortality (626,679 cancer deaths in 2018) in women [155]. Modifiable lifestyle risk factors can play a critical role in cancer prevention [157]. In this respect, several clinical investigations have demonstrated the role of regular, long-term consumption of plant-based diets in reducing breast cancer risk. In a case-control Spanish study [158], adherence to the MedDiet was related to a lower risk [odds ratio (OR) for the top quartile vs. the bottom quartile 0.56 (95% confidence interval (CI) 0.40–0.79)] of breast cancer/overall risk for the top quartile vs. the bottom quartile 0.56 (95% CI 0.40–0.79), whereas adherence to a Western dietary pattern was related to a higher risk [OR for the top vs the bottom quartile 1.46 (95% CI 1.06–2.01)], especially in premenopausal women. Higher consumption of fruits, vegetables, legumes, oily fish, and EVOO significantly decreased the risk of mammary neoplasms, and in particular the most aggressive triple-negative (ER-, PR- and HER2-) subtype [158]. However, stronger evidence for primary chemoprevention of breast malignancies with frequent consumption of EVOO is provided by a prespecified outcome of the PREvención con DIeta MEDiterránea (PREDIMED) trial [159]. After a median follow-up of 4.8 years, on 4282 postmenopausal women, 35 incident cases of breast cancer were identified. The observed incidence rates (per 1000 person-years) of breast cancer were 1.1 for the EVOO-enriched MedDiet group, 1.8 for the nuts-enriched MedDiet group, and 2.9 for the control group [159]. Epidemiological studies assessing the potential health benefits of nutritional interventions have generally investigated the occurrence of individual events as an outcome (e.g., incidence of cancer). Nonetheless, the frequent clustering of cancer and cardiometabolic diseases within the same individual indicates common etiological pathways, in which IR and inflammation are of paramount importance [160,161]. Recently, a large prospective European cohort study [162] investigated the association between lifestyle factors and risk of cancer-cardiometabolic multimorbidity, defined as developing subsequently at least two morbidities, including first cancer at any site (apart from non-melanoma skin cancer), CVD and T2D. Among the 291,778 participants, after a median follow up time of 10.7 years, 22,185 primary cancers, 9016 CVD events and 10,295 newly diagnosed T2D cases were identified. As expected, adiposity was strongly associated with risk of developing a first chronic, IR-related disease, especially T2D, with a hazard ratio (HR) of 2.13 (95% CI, 2.1 to 2.17) per 5% increases in BMI, and more weakly with CVD (HR 1.20 [95% CI, 1.17 to 1.23]) and cancer (HR 1.03 [95% CI, 1.01 to 1.05]) [162]. The 10-years absolute risk estimates for developing multimorbidity ranged between 5 and 17% among cancer patients, and between 20 and 40% among T2D and CVD patients, depending on gender and pre-diagnostic adherence to healthy lifestyle habits. Adherence to the MedDiet reduced the risk of developing CVD and T2D among cancer patients, with a HR of 0.89 (95% CI, 0.81 to 0.97). The beneficial effects of a healthy lifestyle, inclusive of healthy eating habits, in reducing the risk of cancer-cardiometabolic multimorbidity, were particularly marked in patients developing cancer in sites with high 5-year survival rates [162]. These findings are consistent with cross-sectional studies [163] and earlier small cohort investigations [164] that found positive associations between cancer-cardiometabolic multimorbidity and smoking, high alcohol consumption, low physical activity, low fruit and vegetable intake and obesity, supporting World Health Organization (WHO) recommendations for public-health policy to adhere to multiple healthy lifestyle factors for a better prognosis of cancer, CVD and T2D [165]. In addition, an updated meta-analysis of 14 cohort and 18 case-control observational studies has reviewed and summarized the evidence on the association between different dietary patterns and breast cancer risk [166]. The pooled analysis resulted in a positive association between adherence to Western diet and breast malignancies in post-menopausal women, whereas adherence to a “prudent” dietary pattern, that complies well with WHO healthy eating recommendations and rich in vegetables, fruit, fish, poultry and low-fat dairy products, was associated with reduced risk (−23%) of breast malignancies in premenopausal women. However, adjustment for BMI attenuated the magnitude of the correlation between Western diet and risk of breast cancer, suggesting that this eating pattern increases the oncological risk indirectly, through promotion of obesity and consequent adipose tissue dysfunction [7,167]. After menopause, when the ovarian production of estrogens ceases, the adipose tissue remains the major source of circulating estrogens, mainly in the form of estradiol. Obese postmenopausal women have both relatively high serum levels of estradiol and an increased risk of breast cancer, particularly the estrogen/progesterone receptor-dependent (ER+ and/or PR+) subtype [7,168]. As a corollary, adherence to a prudent dietary pattern was significantly associated with a lowered risk of both ER+ and/or PR+ and ER− and PR− breast tumors [166]. Overall, these findings support the role of plant-based diets in regulating not only molecular mechanisms involved in estrogen metabolism, but also other cell signaling transduction pathways and gene expression patterns, which could be either upstream or downstream of the insulin receptor (INSR) signal.

INSR is predominantly involved in the regulation of glucose metabolism in response to insulin and is expressed at high levels in the classical insulin target tissues muscle, adipose tissue and liver, whereas in epithelial cells, it is usually expressed at low levels [169]. INSR overexpression, which occurs in human breast cancer and other epithelial tumors, independently of ER status, remarkably increases the response to circulating insulin, especially when the hormone is abnormally high, as in obesity and T2D. In these circumstances, INSR can exert its oncogenic potential, by directly affecting cell metabolism and/or by synergizing with other oncogenes, with adverse impact on tumor growth and differentiation [169]. The high-mobility group A1 protein (HMGA1) is as a crucial regulator of INSR gene transcription, which directs the assembly of a transcriptionally active multiprotein-DNA complex on the INSR gene promoter [169,170]. As demonstrated by our group [169], the physical and functional cooperation between the transcription factor AP2 and HMGA1 is a fundamental prerequisite for INSR overexpression in neoplastic breast tissues. In addition, breast cancer growth can take advantage of systemic metabolic effects linked to HMGA1-regulated genes in organs distant from the tumor site, reinforcing the close relationship between cancer and abnormal glucose metabolism. HMGA1 regulates both insulin transcription in pancreatic β-cells and expression levels of IGF1-binding proteins involved in glucose disposal in peripheral tissues [170,171], other than a variety of adipokines related to IR and genes relevant for cholesterol biosynthesis [171,172]. Although no data are currently available regarding a potential overlap between cancer and T2D [172], HMGA proteins are also implicated in the multistep processes of tumorigenesis and are susceptible of microRNAs-dependent suppression in normal tissues [173]. Epigenetic silencing of specific microRNAs, such as microRNA-15, microRNA-16, microRNA-26a, microRNA-196a2 and Let-7a, via DNA methylation and histone modification, may lead to increased expression levels of HMGA1 and HMGA2 proteins, likely contributing to tumorigenesis [173]. As concerns possible mechanisms for a chemopreventive role of the MedDiet, several in vitro [174] and in vivo [175,176] studies have described dramatic changes in the transcription of genes and micro-RNAs involved in the pathophysiology of cardiometabolic diseases and cancer, following EVOO interventions. In particular, the acute intake of 50 mL of polyphenol-rich EVOO [175] has been proved to alter the transcriptional regulatory network associated with glucose and lipid metabolism in peripheral blood mononuclear cells, and this paralleled the beneficial effects on glucose and HOMA-IR. Interestingly, the cAMP response element binding protein (CREB) binding protein gene (*CREBBP*) was downregulated following acute EVOO intervention, in both normal weight and obese individuals [175]. CREBBP catalyzes the acetylation of HMGA1 [177] and histone H2B [178], exerting a modulatory effect on HMGA1-linked transcriptional programs in breast cancer cells. Also, given that one of the genes modulated by EVOO is the argonaute RISC catalytic component 2 (AGO2), a master regulator of microRNAs biogenesis and protein synthesis, it is unsurprising that microRNA expression could be also affected by EVOO, with pleiotropic consequences on IR, inflammation and tumorigenesis [175].

## 8. Influence of Polyphenols on Endothelial Dysfunction and Atherosclerosis

Endothelial dysfunction is a key initiating event in the vascular remodeling processes that lead to atherosclerosis, as well as one of the earliest signs of IR [179]. Clinical studies have evidenced an impaired endothelium-dependent dilatation in both T2D patients [180] and related high-risk conditions, including obesity, prediabetes or first-degree familial history of diabetes [181,182,183]. According to current understanding, endothelial dysfunction is associated with oxidative stress and reduced nitric oxide bioavailability, increased anticoagulant properties and platelet aggregation, increased expression of adhesion molecules (i.e., P- and E-selectin, intercellular adhesion molecule-1 (ICAM-1) and leukocyte adhesion molecules [i.e., vascular cell adhesion molecule-1 (VCAM-1), increased secretion of proinflammatory chemokines (monocyte chemotactic protein (MCP-1), and cytokines (i.e., IL-1b, IL-6, IL-8, TNF-a)] [184,185]. A systematic review and meta-analysis [186] reported that a regular consumption of EVOO (approximately between 1 and 50 mg daily) could favorably affect circulating inflammatory biomarkers, known as acute-phase reactants, and the flow-mediated vasodilation, thus contributing to the CVD protective effects of the MedDiet, observed in epidemiological studies [37,38]. Major phenolic compounds within EVOO, such as hydroxytyrosol and oleuropein, can directly target and inhibit the expression of cytokines, chemokines and adhesion molecules, induced by inflammatory stimuli, in human endothelial in vitro systems, via blocking the signaling of nuclear factor-kB (NF-kB), which is a critical regulator of the inflammatory response [187]. However, in a synergistic cooperation with endothelial cells, smooth muscle cells (SMCs) perform also essential functions in sustaining vascular homeostasis, so that their reciprocal interactions may represent novel therapeutic targets for anti-atherogenic interventions [188]. Normally, differentiated SMCs within adult blood vessels proliferate at extremely low rates and produce only a small amount of extracellular matrix components, displaying a contractile phenotype [189]. Nonetheless, under pathophysiological conditions, such as vascular remodeling after endothelial dysfunction, or immunological and mechanical damage, vascular SMCs switch to a de-differentiated, proliferative, and secretory phenotype, which facilitates their ability to migrate to the intima and contribute to the development of atherosclerotic lesions [189].

The architectural transcription factor HMGA1 is a master regulator of the vascular SMCs phenotypic switch that follows vessel wall injury and release of proinflammatory cytokines [6,190,191]. In addition, covering a broad spectrum of mechanistic roles in IR, glucose homeostasis, lipid metabolism and atherogenesis, HMGA1 has been recently proposed as a convincing molecular link between two overlapping pathological traits, such as T2D and CVD [6], other than a potential intervention target for multiple IR-related diseases [172]. In skeletal myocytes, which are the main determinants of insulin sensitivity in humans, saturated fatty acid-induced IR can be rescued by a plant-derived polyphenol through an HMGA1-mediated mechanism [192]. Saturated fatty acids impair insulin biological activity through a kinase independent phosphorylation of the isoform ε of protein kinase C (PKCε) enzyme, which, in its phosphorylated form, migrates to the nuclear region and phosphorylates HMGA1. Phospho-HMGA1 interacts with positively charged histones in heterochromatin regions, reducing its occupation of the *INSR* promoter, and thus, negatively affecting INSR protein expression. In in vitro models of differentiated skeletal myocytes, polyphenol ferulic acid can reduce the activation and nuclear migration of PKCε, induced by saturated fatty acids, and secure the transactivating potential of HMGA1 on *INSR* gene, thus preserving a normal downstream insulin signaling [192]. On the other hand, ferulic acid is also able to attenuate IR in adipocytes of HFD-fed mice, by targeting different pathways, such as those mediated by fetuin-A and NF-kB, ultimately reducing the secretion of proinflammatory cytokines with favorable consequences on the glyco-metabolic status [192]. It is conceivable that other plant-based dietary polyphenols concurrently interact with HMGA1 and NF-kB functions, but their biological activities are likely to be cell type or tissue-dependent, opening a yet unexplored potential for CVD prevention. Figure 4 provides a schematic representation of how polyphenols and saturated fatty acids might influence HMGA1-mediated gene transcription, with divergent effects on glycemic control, tumorigenesis and atherosclerosis.

## 9. MedDiet, EVOO and EVOO-Derived Polyphenols on Hypertension

Hypertension, defined as blood pressure (BP) consistently higher than 130/80 mmHg, according to the most recent American Heart Association guidelines, is the main risk factor for CVD and all-cause mortality [193]. Promoting a healthful lifestyle is a critical first-line strategy for reducing hypertension and its adverse CVD outcomes [193]. However, if targeting whole food combinations in a dietary pattern may have synergistic and cumulative effects on BP over individual foods and nutrients [194], this is even more pronounced for the MedDiet, which offers considerable and secure benefits against the risk of hypertension and CVD. A large cross-sectional study from Tuscany, Central Italy, showed inverse significant associations between specific Mediterranean-based eating patterns, and systolic (SBP) and diastolic blood pressure (DBP) values in non-hypertensive adults [195]. The favorable effects of the MedDiet on BP have been also demonstrated in specific patient groups with multiple CVD risk factors. The milestone interventional PREDIMED trial verified that the MedDiet supplemented with olive oil or nuts could reduce DBP by 22,121.5 mmHg and −0.7 mmHg, respectively, in comparison to a low-fat diet over 4 years in patients with high CVD risk [196]. More recently, the large 1-year multicentric New Dietary Strategies Addressing the Specific Needs of Elderly Population for Healthy Aging in Europe (NU-AGE) trial assessed the effects of a Mediterranean-based dietary intervention, with specific nutritional advices adapted for adults over 65 years of age, counting of high intakes of whole grains, protein (from low-fat dairy, lean meat and fish), low intakes of sodium and vitamin D supplementation (10 μg/day), on BP and vascular stiffness in this special population [39]. Specifically, at 1-year follow-up, SPB decreased by −4.7 mmHg (95% CI, −7.8 to −1.5) in the interventional group, whereas in control group participants, requested to continue with their usual eating habits, a 0.9 mmHg (95% CI, −2.2 to 4.1) SBP increase was observed [39]. Congruous with the observation that in older adults SBP is a more robust CVD risk factor than DBP, and that aging is associated with structural and functional changes in the vascular wall that increase arterial stiffness and SBP, the NU-AGE study found no difference in DPB, following nutritional intervention [39]. In addition, marked gender differences were observed in response to the MedDiet in this study, with reduced SBP in males, and decreased arterial stiffness, but not peripheral BP, in females, which agrees with arterial structure dimorphisms and hormonal influences on BP [39]. However, discrepancies in CVD effects of nutritional interventions, may also arise from gender-specific pharmacokinetics of phenolic compounds contained within foods and dietary supplements [197], which is similar to that of cardiovascular-protective antidiabetic drugs [53]. Although the CVD influence of the MedDiet is mediated by the combination of different foods and healthy lifestyle habits, some specific polyphenol-rich dietary components might be more effective than others in modulating BP and conferring vasoprotection, particularly on a background of IR. EVOO with high polyphenols is probably the most important food from this point of view, being able to improve body composition and reduce BP in obese hypertensive women [198]. The anti-hypertensive actions of EVOO, and EVOO-related functional compounds, appear to be related to an increased endothelial synthesis of nitric oxide (NO), a potent vasodilator [198,199], which is disrupted in carriers of Glu298Asp polymorphism in the endothelial NO synthase (*eNOS*) gene [200] as well as under IR conditions [201], together with a decreased endothelial synthesis of the vasoconstrictor endothelin (ET-1) [199]. The NO/ET-1 unbalance, which alters the vascular tone and drives hypertension, can be induced by increased plasma levels of FFAs and glucose, two well-known features of IR states [201]. In endothelial in vitro models, hydroxytyrosol and the total polyphenolic extracts from EVOO partially reversed the intracellular NO levels that were reduced by elevated medium concentrations of glucose and FFAs [199]. These effects were related to a positive modulation of insulin signaling, that, via PI3K/Akt, is considered a critical regulator of eNOS phosphorylation and activity [199].

## 10. MedDiet, EVOO and EVOO-Derived Polyphenols on Lipid Abnormalities

There is compelling longstanding evidence that dyslipidemia is a major modifiable risk factor for the development and progression of CVD [202]. It has been also recognized that, irrespective of T2D, a complex interplay between IR and plasma lipid abnormalities exists, and this could be a novel target for cardioprotective interventions [203]. Indeed, insulin-sensitizing dietary approaches have a significant impact on lipid metabolism and CVD risk and should be recommended for cardiovascular prevention [202]. In particular, the PREDIMED trial demonstrated beneficial effects of the EVOO-enriched MedDiet on primary prevention of CVD in at-risk individuals, which appears to be related, at least in part, to an improved and less atherogenic lipid profile, in terms of resistance against oxidation, size and cytotoxicity of low-density lipoprotein (LDL) particles [204]. In this regard, it has been highlighted that oxidized LDL particles might play a role in atherosclerosis onset and progression, given their ability to induce endothelial dysfunction and macrophagic transformation into foam cells, following phagocytosis of LDLs [204,205]. Also, compared to normal size LDL, small LDL particles possess decreased affinity for LDL receptors and have a longer permanence in the bloodstream, where are easily oxidized, becoming more capable to traverse the endothelial barrier [206]. Considering that high concentrations of small LDL lipoproteins are associated with a greater incidence of atherosclerosis and CVD events [207], it is plausible that the cardioprotective effects of EVOO are related to increased LDL size and resistance against oxidation, which are all consequences of its polyphenolic content and antioxidant capacity [208]. Beside PREDIMED, several studies have shown a decrease in the ability of LDL to be oxidized after consumption of polyphenol-rich EVOO [209,210], although the true contribution of such modifications to CVD risk reduction remains elusive [208]. In addition, randomized controlled trials have reported a dose-dependent increment in high-density lipoprotein (HDL) levels with EVOO [211,212], and these, contrary to LDL species, are strongly inversely associated with CVD [213]. The antiatherogenic functions of high-density lipoproteins (HDLs) have been investigated in terms of both plasma concentrations and cholesterol efflux capacity (CEC), the latter expressing the ex vivo ability of an individual’s HDL to promote cholesterol efflux from macrophages in cell culture [213]. In this regard, when compared to a low-fat diet, adherence to either EVOO-enriched or nut-enriched MedDiet was associated with a trend of decrement in HDL levels, along with a significant increase in CEC [214], which was linked to a reduced risk of first CVD events [214]. Given the similar effects of EVOO and nuts supplementations on HDL quality, it is quite possible that this is an added value of the MedDiet, that focuses on whole foods and nutrients combination. More recently, a cross-sectional analysis of the baseline data of overweight and obese participants enrolled in the PREDIMED-Plus trial emphasized the effects of dietary polyphenol intake and polyphenol subclasses on individual components of the metabolic syndrome, including, among others, plasma lipid abnormalities [215]. Positive associations were found between HDL levels and all polyphenol classes except for phenolic acid and lignan intake [215]. Similar positive associations between HDL and total polyphenol intake were also found in patients with T2D [216], and elderly non-diabetic patients at high CVD risk [217], reinforcing the protective or therapeutic effects of the MedDiet on lipid abnormalities under different IR-related situations.

## 11. MedDiet and Management of Chronic Kidney Disease (CKD)

CKD, a condition characterized by a gradual loss of kidney function until severe insufficiency and need of replacement therapy, is associated with a high risk of CVD and CVD-related mortality, even in non-diabetic patients [218]. The strong relationship between CKD and CVD can be explained with the notion of shared inflammatory and glycometabolic abnormalities, frequently related to IR, as mutual risk factors. In the last decades, insulin action defects have been reported in the early stages of CKD [219,220] as well as in CKD progressors [221], and it has been proposed that IR can promote kidney damage through hemodynamic mechanisms, including a sympathetic nervous system overactivity, increased sodium retention, glomerular hyperfiltration and elevated vascular permeability [222]. CKD is also considered a major risk factor for IR development, given the unbalance of circulating pro-inflammatory cytokines and adipokines that follows the reduction in kidney function. In particular, leptin accumulates to a larger extent than adiponectin in CKD, and this highly abnormal ratio may favor the onset of IR and metabolic disorders [223,224]. However, recently, the Chronic Renal Insufficiency Cohort (CRIC) study [225] questioned the causal link between IR and progression of CKD in absence of diabetes. In this large prospective observational cohort study, enrolling 3939 non-diabetic patients with CKD from the United States, there were no significant association between baseline HOMA-IR and CKD progression, atherosclerotic CVD event, or all-cause mortality during 10 years of follow-up [225]. On the other hand, previous reports on CKD demonstrated a more severe reduction of kidney function in patients with IR [226,227], so that the available evidence in this context remains inconclusive. Also, it cannot be excluded that different dietary management could have confounded the results. In this regard, emerging evidence suggests that fruit and vegetable-rich diets, such as the MedDiet, may be helpful to delay CKD progression and prevent complications [228]. Based on these considerations, the MedDiet has been recently promoted by the European Renal Association–European Dialysis Transplant Association (ERA-EDTA) as the most appropriate choice for the nutritional management of CKD [229]. Protein intake in the MedDiet is very close to that of traditional controlled protein diets for CKD (~0.8 g/kg/day). Also, in Mediterranean Countries, red meat and processed meats are consumed less than vegetables, fish and white meat, and this may convey a lower amount of dietary sodium, phosphate and potassium, which represent a major concern in patients with CKD due to an intrinsic risk of water-electrolyte decompensation [229]. A lower consumption of dairy, red meat and meat processed products would also reduce the proinflammatory saturated fatty acid intake, possibly ameliorating the atherogenic lipid levels and systolic blood pressure [230], where the abundant supply of dietary fiber would mitigate the systemic inflammatory milieu. Indeed, the Third National Health and Nutrition Examination Survey (NHANES III) showed that the total dietary fiber intake was low in most cohort participants (median 14.5 g of fiber/day), and higher dietary fiber consumption was associated with lower serum levels of C reactive protein, a widely used clinical marker of inflammation, and lower risk of mortality in the subpopulation with CKD. Specifically, each 10 g/day increase in total dietary fiber reduced the risk of mortality by 17% in this population [231]. However, as underlined by ERA-EDTA itself, the high consumption of fresh fruits and vegetables typical of the Mediterranean traditions and culinarian style, enhances the risk of hyperkalemia in advanced stages of CKD, so that low potassium alternatives or boiling cooking methods should be still cautiously considered when adapting the MedDiet to patients with reduced kidney function [229].

Clinical and preclinical evidences on the role of the MedDiet and its functional compounds against IR and IR-related diseases, which have been discussed in this review, are reassumed in Table 1 and Table 2, respectively.

## 12. Limitations and Future Research Perspectives

Although there is a growing range of studies evaluating the anti-inflammatory, anti-obesity, anti-diabetic and anti-cancer properties of the MedDiet functional components, schematically illustrated in Figure 5, the present knowledge about their role in the modulation of insulin action and signaling pathways is still limited. Further studies, with particular emphasis on in vivo approaches, are required to determine their therapeutic potential in humans and to provide a better understanding of their biological activities at different tissue levels. Also, given the recent characterization of a large number of diverse bioactive compounds from plant-based natural products, which would challenge the feasibility of a large-scale in vitro testing for the identification of potential health effects [232], novel in silico approaches have been proposed. Methods such as molecular docking, virtual screening, and molecular dynamics simulation, can provide a critical basis for understanding the complex interaction of these compounds with cellular enzymes and regulatory molecules [233]. While isolation of natural compounds directly from the plant species is acceptable when only small to moderate amounts are required, sustainable synthetic routes for resupply need to be considered once the healthcare demands increase. However, despite representing a novel treatment paradigm for IR and IR-related diseases, polyphenol-based synthetic supplements may have several drawbacks [232]. Often polyphenols act synergistically with other nutrients to influence glucose homeostasis and cellular processes, so it is uncertain whether isolated supplements would have the maximum effect without eating a variety of whole foods, as in the MedDiet eating pattern [232]. Furthermore, the in vivo safety of high doses of pure compounds (over 100 times higher than the natural occurrence in some cases) is unlikely to be feasibly extrapolated from in vitro mechanistic research, with several studies reporting the increased risk of hepatotoxicity [234], and others highlighting the increased risk of stroke and premature death [235], and even carcinogenesis [236]. As a consequence, more investigations are needed to establish which dosages of the isolated polyphenol compounds are safe and effective for clinical use, testing first the concentrations close to those present in humans as a result of a healthy dietary intake [237]. Finally, even for some of the most promising dietary supplements explored in this review, longer follow-up periods and larger sample sizes are essential to assess the interventions’ outcomes in the long run [119,120,121,123,138].

## 13. Conclusions

Adherence to the MedDiet affords protection from hard curative IR-related diseases such as obesity, T2D, NAFLD, cognitive impairment, CVD, CKD, PCOS and breast cancer. Given the growing evidence regarding the intake of particular functional components of the MedDiet on the modulation of disease-specific pathognomonic traits, in the present narrative review we aimed at providing instances of mechanistic explanation for the observed clinical phenomena from a compound-centric perspective. Interestingly, according to nutrigenomic studies, some of the health benefits can relate to the ability of several plant-based polyphenols to positively affect gene transcription patterns, in which HMGA1 acts as a critical regulator, opening new avenues for researchers. The identification of nutrients, regulating molecular pathways particularly relevant for glucose homeostasis, cognitive functions, tumorigenesis or atherogenesis, in either an individual or cooperative manner, may contribute to the formulation of functional food-based dietary guidelines for managing the clinical spectrum of IR, reducing the need for pharmacological interventions, with a favorable impact on healthcare. However, even if older people with mild cognitive decline may take advantage of nutritional supplements for correcting dietary inadequacies and deficiencies in a practical manner, uncertainties surround the likelihood of metabolic benefits and safety of unsaturated fatty acid-enriched diets and polyphenol-based synthetic supplements at a general population scale. On the other hand, considering its merits and sustainability, the adoption of the MedDiet as a healthy eating behavior should be further encouraged by public nutritional policies.

## Figures and Tables

**Figure 1 nutrients-12-01066-f001:**
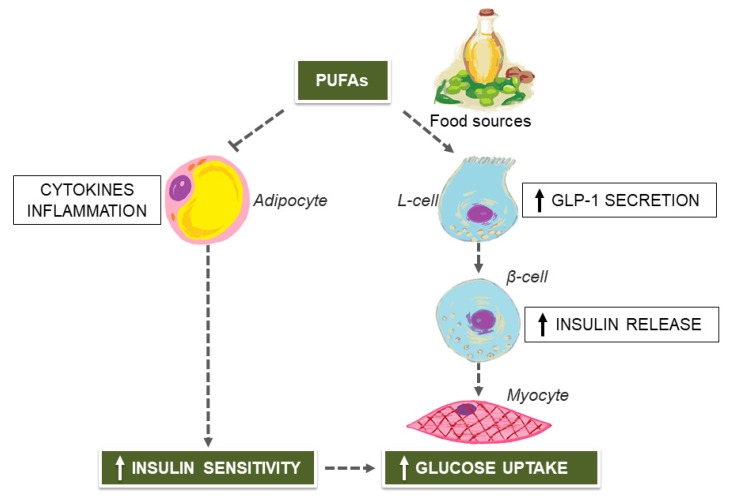
Polyunsaturated fatty acids (PUFAs) from olive oil, nuts, and seeds and the amelioration of glycemic escursions. GLP-1, glucagon-like peptide 1.

**Figure 2 nutrients-12-01066-f002:**
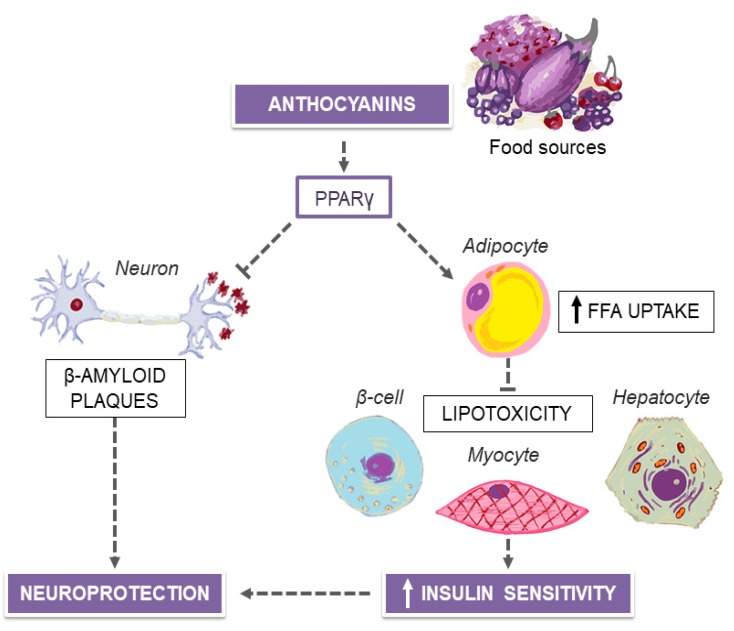
Anthocyanins from purple-colored vegetables and fruits and the mechanisms for neuroprotection. PPARγ, peroxisome proliferator-activated receptor γ; FFA, free fatty acids.

**Figure 3 nutrients-12-01066-f003:**
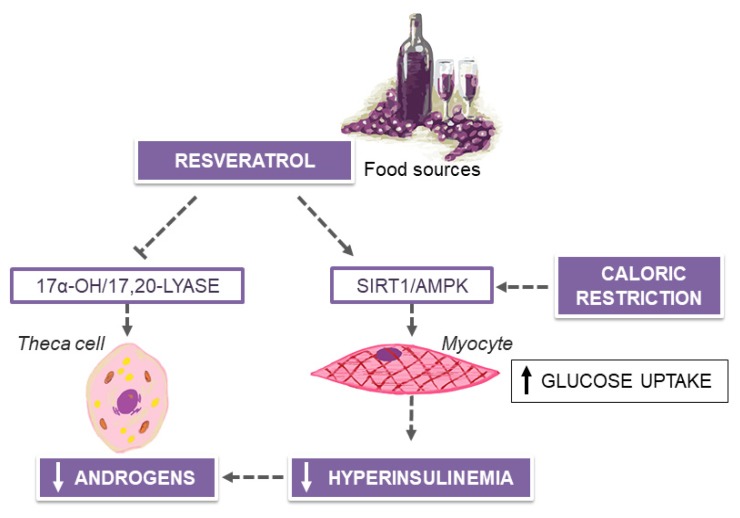
Resveratrol from grapes, berries and wine and the amelioration of hyperandrogenism in polycystic ovary syndrome (PCOS). SIRT1, sirtuin 1; AMPK, AMP-activated protein kinase.

**Figure 4 nutrients-12-01066-f004:**
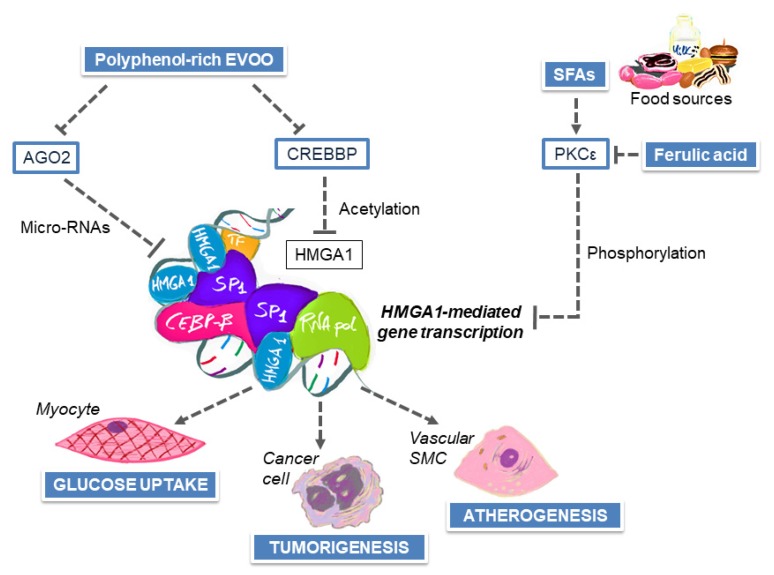
Dietary modulation of HMGA1 functions in the maintenance of glycemic homeostasis, tumorigenesis and atherosclerosis. EVOO, extra virgin olive oil; SFAs, saturated fatty acids; AGO2, argonaute RISC catalytic component 2; CREBBP, CREB (cAMP response element binding protein) binding protein; PKCε, protein kinase C isoform ε; HMGA1, high-mobility group A1 protein.

**Figure 5 nutrients-12-01066-f005:**
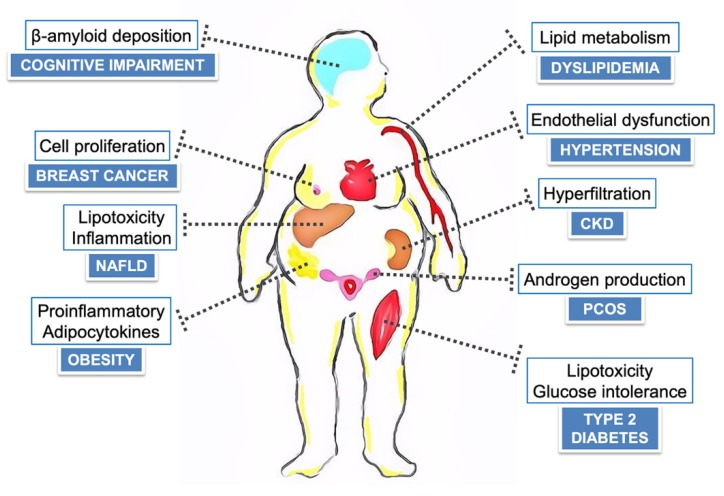
Beneficial effects of the MedDiet on IR-related traits: summary. NAFLD, non-alcoholic fatty liver disease; CKD, chronic kidney disease; PCOS, polycystic ovary syndrome.

**Table 1 nutrients-12-01066-t001:** Summary of clinical studies assessing the efficacy of MedDiet and nutritional supplements against IR and IR-related diseases.

Design	Subject	Diet/Supplement	Main Findings	Ref.
Prospective cohort study	1302 patients with CVD	MedDiet	Higher adherence to the MedDiet was associated with a lower all-cause and CVD related-mortality rate (−27% and −31%, respectively) over 3.78 years of follow up.	[37]
Prospective cohort study	22,043 adults free of cancer, T2D or CVD	MedDiet	Higher adherence to the MedDiet was associated with a lower all-cause, cancer and CVD related-mortality rate (−25%, −24% and −33%, respectively) over 44 months of follow up.	[38]
RCT	1294 elderly adults	MedDiet adapted for the adults ≥65 years of age	After 1-year follow up, adherence to the MedDiet was associated with decreased levels of SPB (−4.7 mmHg), whereas in control group participants, requested to continue with their usual eating habits, SBP increased by 0.9 mmHg. Differences in SBP between groups were significant for males, but not for females.	[39]
Prospective cohort study	12,168 middle-aged adults without CVD	Plant-based diets	Higher adherence to an overall plant-based diet, or a provegetarian diet, was associated with a lower risk of CVD-related and all-cause mortality (−19% and −11%, respectively) over 19 years of follow up.	[42]
Metanalysis of RCTs	1460 patients with T2D	MUFA-enriched diets vs. CHO-enriched diets	High-MUFA diets were associated with significant reductions in fasting plasma glucose (WMD −0.57 mmol/L), triglycerides (−0.31 mmol/L) body weight (−1.56 Kg), and SBP (−2.31 mmHg) along with significant increases in HDL cholesterol (0.06 mmol/L) when compared to high-CHO diets.	[43]
Prospective cohort study	13,380 adults without T2D	MedDiet	Higher adherence to the MedDiet was associated with lower incidence of T2D over 4.4 years of follow up. A two-point increase in the adherence score was associated with a 35% relative risk reduction of developing T2D.	[44]
RCT	418 middle-aged and elderly adults without T2D	EVOO-enriched (1 L/week) MedDiet vs. nut-enriched (30 g/day) MedDiet vs. low-fat diet	Adherence to both EVOO-enriched and nut-enriched MedDiets were associated with lower incidence of T2D over a median 4-year follow up (HR of T2D 0.49 and 0.48, respectively) when compared to low-fat diet.	[45]
Metanalysis of RCTs	121,070 patients with T2D or at risk for T2D	Individual or total PUFA supplements	PUFA supplements had little or no effect on likelihood of T2D diagnosis (RR 1.00) or measures of glucose metabolism and IR (mean differences in HbA1c −0.02%; plasma glucose 0.04 mmol/L; fasting insulin 1.02; HOMA-IR 0.06).	[54]
Metanalysis of prospective cohort studies	312,015 adults without T2D	High (33.2 to 1452.3 mg/day) vs. low (8.9 to 501.8 mg/day) total flavonoid intake	High total flavonoid intake was associated with decreased risk (−11%) of developing T2D during 4–28 years of follow-up, when compared to a low intake. Each 300 mg/day increment in flavonoids consumption was associated with 5% reduction in T2D risk. The protective effect was significant for anthocyanidins, flavan-3-ols, flavonols, and isoflavones.	[58]
Cross-sectional	58 patients with NAFLD	MedDiet	Higher adherence to the MedDiet was negatively correlated to serum liver enzymes, fasting insulin, HOMA-IR and NAFLD severity, and positively correlated to serum adiponectin levels. Patients with NASH exhibited lower adherence to the MedDiet compared to those with simple steatosis.	[76]
Crossover RCT	12 patients with NAFLD	MedDiet vs. low-fat high CHO diet	A 6-week MedDiet intervention enhanced the relative reduction in hepatic steatosis (−39 ± 4%) when compared to a low-fat high CHO diet (−7 ± 3%). Insulin sensitivity, assessed by hyperinsulinemic clamp, improved significantly with the MedDiet intervention, but not with the control diet.	[77]
RCT	98 patients with moderate-severe NAFLD	Low-glycemic index MedDiet	Adherence to a low-glycemic index MedDiet significantly reduced the NAFLD score (−4.14) within 6 months.	[78]
Prospective intervention study	44 patients with liver steatosis	MedDiet	Adherence to the MedDiet was associated with significant amelioration of clinical, biochemical, and inflammatory biomarkers of NAFLD after 24 weeks. *STAT3* rs2293152 G-carriers experienced more beneficial changes at the end of the dietary intervention.	[79]
Prospective cohort study	70 cognitively normal middle-aged adults	MedDiet	Lower adherence to the MedDiet was associated with progressive AD biomarkers abnormalities, including lower FDG-PET glucose metabolism, and higher β amyloid load in AD-affected brain regions.	[96]
Cross-sectional	4447 adults without dementia or cerebrovascular disease	High quality vs. low quality diets	High quality diets, with high consumption of vegetables, fruit, whole grains, nuts, dairy, and fish and low intake of sugar-containing beverages, were related to larger brain volumes, gray matter volumes, white matter volumes, and hippocampal volumes.	[97]
RCT	49 elderly patients with mild to moderate AD	Anthocyanin-rich cherry juice (200 mL/day)	Twelve-week intervention with cherry juice was associated with significant improvements in verbal fluency, short-term memory, long-term memory and SBP levels when compared to the control group. Markers of inflammation (CRP and IL-6) were unchanged.	[119]
RCT	12 elderly patients with mild cognitive impairment	Flavonoids-rich concord grape juice (6–9 mL/Kg/day)	Twelve-week intervention with flavonoids-rich concord grape juice was associated with improvement in verbal learning and non-significant enhancement of verbal and spatial recall. A small increase in fasting insulin was also observed.	[120]
RCT	44 elderly patients with AD	Coconut oil-enriched (40 mL/day) MedDiet	After 21 days of intervention with coconut oil-enriched MedDiet, improvements in episodic, temporal orientation, and semantic memory were observed, and these were more pronounced in women with mild-moderate disease.	[123]
Cross-sectional	112 treatment-naïve women with PCOS	MedDiet	PCOS women showed higher testosterone levels, Ferriman–Gallwey/hirsutism score, fasting insulin, fasting glucose levels and HOMA-IR when compared with control healthy women. Despite no differences in energy intake, PCOS women consumed less EVOO, legumes, fish/seafood, and nuts with a higher quantity of simple carbohydrate, total fat, SFA when compared to controls. In PCOS women, adherence to the MedDiet was negatively associated with testosterone levels.	[135]
Prospective cohort study	259 healthy premenopausal women	MedDiet	Adherence to the MedDiet was associated with decreased plasma biomarkers of lipoperoxidation and oxidative stress, such as 8-iso-PGF2α and 9-HODE, and increased levels of ascorbic acid.	[137]
RCT	34 women with PCOS	1500 mg/day of oral micronized trans-resveratrol	After 3 months, resveratrol treatment led to a significant decrease of total testosterone, dehydroepiandrosterone sulfate and fasting insulin (−23.1%, −22.2%, and −31.8%, respectively), along with an increase of the Insulin Sensitivity Index (66.3%), when compared to placebo.	[138]
RCT	50 patients with NAFLD	500 mg/day of resveratrol	After 12 weeks, resveratrol supplementation reduced liver enzymes and steatosis significantly more than placebo. BP, IR markers or body weight remained unchanged.	[143]
Metanalysis of RCTs	283 patients with T2D	8–3000 mg/day of resveratrol	Treatment with resveratrol significantly improved fasting plasma glucose (−0.29 mmol), insulin levels (−0.64 U/mL), HOMA-IR (−0.52), SBP and DBP (−0.58 and −0.43 mmHg, respectively) when compared to placebo. Subgroup analysis revealed that resveratrol supplementation doses ≥ 100 mg/day were associated with more favorable results.	[151]
Case-control	1017 women newly diagnosed with breast cancer, and 1017 healthy matched controls	MedDiet	Compared with controls, women with incident breast cancer were more likely to adhere to a high-energy Western diet. Adherence to the MedDiet was associated with decreased risk of breast cancer (OR 0.56), that was more pronounced for triple-negative tumors (OR 0.32), whereas adherence to the Western diet was related to a higher risk (OR 1.46), especially in premenopausal women.	[158]
RCT	4152 women at high CVD risk without breast cancer	EVOO-enriched (1 L/week) MedDiet vs. nut-enriched (30 g/day) MedDiet vs. low-fat diet	After a median follow up of 4.8 years, breast cancer rates (per 1000 person-years) were 1.1 for the EVOO-enriched MedDiet, 1.8 for the nut-enriched MedDiet, and 2.9 for the control low-fat diet, with adjusted HR for the EVOO-enriched and nut-enriched MediDiets of 0.32 and 0.59, respectively.	[159]
Prospective cohort study	291,778 middle-aged adults free of cancer, T2D, CVD	MedDiet	After a median cohort follow up of 10.7 years, 22,185 primary cancers, 9016 CVD events and 10,295 new cases of T2D were identified. Adherence to the MedDiet significantly reduced the risk of developing CVD and T2D multimorbidity among cancer patients, with a HR of 0.89.	[162]
Metanalysis of observational studies	43,285 women with breast cancer	Prudent diet vs. western diet	Adherence to a prudent diet was associated with 23% reduced risk of breast cancer in premenopausal women, irrespective of the hormonal receptor status. Adherence to the Western diet resulted in 20% increased risk of ER+ and/or PR+ breast cancer in post-menopausal women.	[166]
RCT	7447 middle-aged and elderly patients at high CVD risk	EVOO-enriched (1 L/week) MedDiet *vs.* nut-enriched (30 g/day) MedDiet vs. low-fat diet	After a median follow up of 4 years, the percentage of participants with controlled BP increased in all groups. However, adherence to EVOO-enriched and nut-enriched MedDiets was associated with significantly lower DBP values (–1.53 and –0.65 mmHg, respectively) when compared to the control low-fat diet.	[196]
RCT	41 overweight young adult women	EVOO-enriched (25 mL/day) vs. soybean oil-enriched (25 mL/day) energy restricted normal fat diet	After 9 weeks, adherence to the EVOO-enriched diet was associated with higher fat loss (−2.4 vs. −1.3 Kg) and reduced DBP levels (−5.1 vs. 0.3 mmHg), than the control diet. There was also a trend of reduction for the proinflammatory IL-1β with EVOO.	[198]
RCT	210 patients at high CVD risk	EVOO-enriched (1 L/week) MedDiet vs. nut-enriched (30 g/day) MedDiet vs. low-fat diet	After 1 year of follow up, adherence to the EVOO-enriched MedDiet significantly increased LDL resistance against oxidation (+6.46%) and estimated LDL particle size (+3.06%), with respect to the low-fat-diet. Adherence to the nut-enriched MedDiet was not associated with changes in LDL traits.	[204]
RCT	296 patients at high CVD risk	EVOO-enriched (1 L/week) MedDiet vs. nut-enriched (30 g/day) MedDiet vs. low-fat diet	After 1 year follow up, adherence to both the EVOO-enriched and nut-enriched MedDiets significantly increased CEC relative to baseline, by improving HDL oxidative status and composition.	[214]
Metanalysis of prospective cohort studies	15,285 patients with CKD	Plant-based healthy diets	Adherence to plant-based healthy diets, including MedDiet, were consistently associated with lower all-cause mortality (−27%) when compared to other dietary patterns. There was no statistically significant association between adherence to healthy diets and risk of ESRD.	[228]

IR, insulin resistance; CVD, cardiovascular disease; T2D, type 2 diabetes mellitus; RCT, randomized controlled trial; MUFA, monounsaturated fatty acid; CHO, high carbohydrate; WMD, weighted mean difference; HDL, high-density lipoprotein; LDL, low-density lipoprotein; EVOO, extra virgin olive oil; HR, hazard ratio; PUFA, polyunsaturated fatty acid; RR, relative risk; HbA1c, glycated hemoglobin; HOMA-IR, homeostatic model assessment for insulin resistance; NAFLD, non-alcoholic fatty liver disease; NASH, non-alcoholic steatohepatitis; STAT3, signal transducer and activator of transcription 3; FDG-PET, fluorodeoxyglucose-positron emission tomography; AD, Alzheimer disease; SBP, systolic blood pressure; DBP, diastolic blood pressure; CRP, C reactive protein; IL-6, interleukin 6; PCOS, polycystic ovary syndrome; SFA, saturated fatty acid; OR, odds ratio; IL-1β, interleukin 1β; ER, estrogen receptor; PR, progesterone receptor; CKD, chronic kidney disease; ESRD, end-stage renal disease; CEC, cholesterol efflux capacity.

**Table 2 nutrients-12-01066-t002:** Summary of preclinical studies exploring the bioactivity potential of Mediterranean nutrients against IR.

Design	Subject	Nutrient (Dose)	Main Findings	Ref.
In vivo	HFD-fed mice	PUFA-enriched oil (30% of total energy)	Restoration of HFD-induced glucose intolerance, vascular dysfunction and hypercholesterolemia, via enhanced Akt/PKB phosphorylation.	[46]
In vivo	Obese Zucker rats	PUFA-enriched oil (10% of total energy)	Restoration of obesity-induced glucose intolerance via enhanced Akt/PKB phosphorylation and mitochondria bioenergetics.	[47]
In vitro	L6 rat skeletal muscle cells	MUFAs and PUFAs (200–700 μM)	Dose-dependent enhancement and prolongation of insulin-induced Akt/PKB and ERK1/2 phosphorylation, via repression of PP2A activity.	[48]
In vitro	C2C12 mouse skeletal muscle cells; L6 rat skeletal muscle cells	Quercetin and quercetin 3-O-glycosides (25–100 μM)	Enhanced muscular glucose uptake (38%–59%) in the absence of insulin via activation of the AMPK pathway and translocation of GLUT4.	[59,60]
In vitro	Yeast α-glucosidase	Flavonoids (0–200 μM)	Strong inhibition of α-glucosidase activity (IC50 ≤ 200 μM).	[62]
In vivo	HFD-fed mice	Oleacein (20 mg/Kg/day)	Prevention of HFD-induced adiposity, hyperglycemia, hyperinsulinemia, hyperlipidemia and liver pathology via reduced FAS, SREBP-1, ERK, and p-ERK liver protein levels. No signs of oleacein-induced organ toxicity.	[85]
In vivo; in vitro	HFD-fed mice; 3T3-L1 mouse preadipocytes	Oleacein (20 mg/Kg/day; 0–100 µM)	In vivo prevention of HFD-induced increase of adipocyte size, adipose tissue inflammation and fibrosis, via reduction of PPARγ and SREBP-1 protein expression; enhancement of adiponectin production in adipose tissue and increased expression of GLUT4 in skeletal muscle cells. In vitro dose-dependent prevention of lipid droplets accumulation during adipocyte differentiation via inhibition of FAS and PPARγ.	[87]
In vivo; in vitro	APP/PS1 mice; SH-SY5Y human neuroblastoma cells	Cyanidin-3-O-glucopyranoside (5 mg/Kg/day; 25 μM)	In vivo amelioration of object recognition, spatial memory, behavioral abnormalities and glucose intolerance. In vitro cytoprotective effects against amyloid β-induced toxicity via PPARγ upregulation.	[107]
In vitro	Rat ovarian theca-interstitial cells	Resveratrol (30–100 µM)	Dose-dependent inhibition of cell growth and cell viability; counteraction of insulin-induced pro-proliferative and anti-apoptotic effects.	[139]
In vitro	Rat ovarian theca-interstitial cells	Resveratrol (1–10 µM)	Inhibition of androstenedione and androsterone production (−78% and −74%, respectively), via Akt/PKB signaling pathway.	[140]
In vitro	Human embryonic kidney 293T cells; Human epithelial HeLa cells; HUVEC human endothelial cells; 3T3-L1 mouse preadipocytes; CHO hamster ovarian cells	Resveratrol (50–100 μM)	Cell-specific activation of AMPK, that can be linked to ATP synthase inhibition (energy restriction-sensitive), SIRT1-LKB1 stimulation, or both mechanisms.	[148]
In vivo	HFD-fed mice	Resveratrol (5.2–22.4 mg/Kg/day)	Longer lifespan, increased insulin sensitivity, reduced IGF-1 levels, increased liver AMPK activity, decreased liver, heart and aorta pathology, increased mitochondrial number, and improved motor function under chronic high fat hypernutrition.	[149]
In vitro	HFD-fed Rats; L6 rat skeletal muscle cells	Ferulic acid (0.6 mg/Kg/day; 2–20 µg/mL)	In vivo prevention of IR in adipose tissue, stimulated by SFA oversupply, via fetuin-A downregulation. In vitro dose-dependent improvement of SFA-induced muscular IR via inhibition of PKCε phosphorylation and restoration of HMGA1-mediated transcription of the *INSR* gene.	[192]
In vitro	ECV304 human endothelial cells	Hydroxytyrosol and EVOO total polyphenol extract (10 µM)	Prevention of NO reduction and ET-1 synthesis, induced by elevated glucose and FFA concentrations, via PI3K/Akt modulation.	[199]

IR, insulin resistance; HFD, high-fat diet; PUFA, polyunsaturated fatty acid; MUFA, monounsaturated fatty acid; Akt/PKB, protein kinase B; ERK, extracellular signal-regulated kinase; PP2A, protein phosphatase 2A; APP/PS1, double transgenic mutant human amyloid precursor protein/presenilin 1; PPARγ, peroxisome proliferator-activated receptor γ; AMPK, AMP-activated protein kinase; SIRT1, sirtuin 1; LKB1, liver kinase B1; IGF-1, insulin-like growth factor 1; SFA, saturated fatty acid; PKCε, protein kinase C isoform ε; HMGA1, high-mobility group A1 protein; INSR, insulin receptor; FAS, fatty acid synthase; SREBP-1, sterol regulatory element-binding transcription factor-1; GLUT4, glucose transporter 4; EVOO, extra virgin olive oil; NO, nitric oxide; ET-1, endothelin; FFA, free fatty acid; PI3K, phosphoinositide 3 kinase.

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
