# Peer review of "Mediterranean Diet Nutrients to Turn the Tide against Insulin Resistance and Related Diseases"

_nutrients, 2020, doi:10.3390/nu12041066_

Round 1

Reviewer 1 Report

We really believe that this review is very interesting and necessary right now to the scientific community. Papers updated and commented very well throughout the article. Perhaps, the only thing that the authors could improve is the distribution of the sections because they focus only in one nutrient/group of nutrients for each type of pathology for each type of pathology, but many times it is the synergy between various groups that confers the Benefit

MAJOR points

1.- we suggest to the authors to separate in vitro/in vivo trials, clinical trials and epidemiological resultas within each section (may be introduce new sub-sections), becuase sometimes are confusing.

2.- We suggest making a "summary" figure of all the parts. If it is argued that the Mediterranean diet is a pattern made up of different foods, with different cardiometabolic benefits, it is logical to integrate it into a section / figure that could be included as a conclusion.

MINOR POINTS

1.- we suggest abbreviating Mediterranean diet as MedDiet since it is widely used

2.- Make the arrows bigger. They do not look so well.

3.- Use a different color / box type nomenclature for effects that is different from the genes in the figures. PPARy different from Lipotoxicity in Figure 2.

4.- Indicate the "food sources" in the figure footer. E.g. Figure 1. Polyunsaturated fatty acids (PUFAs) from olive oil and nuts, and the amelioration (..)

5.- In the Figure 3, define PCOS

6.- In the figure 4, they have not defined CREB and the names of the HMGA1 subunits do not read well.

Author Response

Authors: We thank the reviewer for the careful reading of the manuscript and his/her constructive remarks. We have accepted most of the comments to improve our manuscript. Please find below a detailed point-by-point response (reviewers' comments in black, our replies in blue italic).

REVIEWER 1

We really believe that this review is very interesting and necessary right now to the scientific community. Papers updated and commented very well throughout the article. Perhaps, the only thing that the authors could improve is the distribution of the sections because they focus only in one nutrient/group of nutrients for each type of pathology for each type of pathology, but many times it is the synergy between various groups that confers the Benefit

Authors: We thank the reviewer for the appreciation. We have now remarked the critical role of a synergistic cooperation between different nutrients and foods in determining the health benefits of the Mediterranean diet in lines 543-546 (section 8 “MedDiet, EVOO and EVOO-derived polyphenols on hypertension”) lines 611-615 (section 9 “MedDiet, EVOO and EVOO-derived polyphenols on lipid abnormalities), lines 697-700 (section 11“Limitations and future research perspectives).

MAJOR points

1.- we suggest to the authors to separate in vitro/in vivo trials, clinical trials and epidemiological resultas within each section (may be introduce new sub-sections), becuase sometimes are confusing.

Authors: Given the insertion of four novel sections (namely “3. MedDiet flavonoids for preventing T2D”, “8. MedDiet, EVOO and EVOO-derived polyphenols on hypertension”, “9. MedDiet, EVOO and EVOO-derived polyphenols on lipid abnormalities”, “10. MedDiet for managing CKD”) in the revised version of the manuscript, we have now added two tables (pages 16-21) to reassume the design and main findings of clinical and preclinical investigations, respectively.

2.- We suggest making a "summary" figure of all the parts. If it is argued that the Mediterranean diet is a pattern made up of different foods, with different cardiometabolic benefits, it is logical to integrate it into a section / figure that could be included as a conclusion.

Authors: We have now added a summary figure of the beneficial effects of the Mediterranean diet on many pathological traits related to insulin resistance.

MINOR POINTS

1.- we suggest abbreviating Mediterranean diet as MedDiet since it is widely used

Authors: The “MedDiet” abbreviation has been now used in the revised version of the manuscript.

2.- Make the arrows bigger. They do not look so well.

Authors: In all figures, we have now increased the arrow size to 3 pt.

3.- Use a different color / box type nomenclature for effects that is different from the genes in the figures. PPARy different from Lipotoxicity in Figure 2.

Authors: We have now changed the shape outline of the following text boxes: PPARy (Figure 2); SIRT1/AMPK and 17αOH/17-20 LYASE (Figure 3); AGO2, CREBBP and PKCε (Figure 4).

4.- Indicate the "food sources" in the figure footer. E.g. Figure 1. Polyunsaturated fatty acids (PUFAs) from olive oil and nuts, and the amelioration (..)

Authors: Food sources are now indicated in figure footers.

5.- In the Figure 3, define PCOS

Authors: We have how defined PCOS and all missed abbreviations in figure footers, table footers and manuscript text, where appropriate.

6.- In the figure 4, they have not defined CREB and the names of the HMGA1 subunits do not read well.

Authors: We have now defined CREB as “cAMP response element binding protein”, and slightly changed the size and colors of the enhanceosome in figure 4.

Reviewer 2 Report

The aim of the review is to highlight the role of the Mediterranean diet and selected nutritional supplements on insulin resistance and related diseases.

Major:

  • The role of polyphenols on glucose homeostasis is glossed over in this review. I suggest to insert a paragraph about the potential role of polyphenols in the prevention of type 2 diabetes mellitus. In the last few years several molecular and clinical studies have been conducted to evaluate the effects of flavonoids and their major food source on insulin resistance (IR). Please review this topic.
  • IR is associated with increased risk for chronic kidney disease (CKD) in non-diabetic patients. Does the Mediterranean diet and its bioactive compounds have an effect on this IR-related disease?
  • In the metabolic syndrome, IR is related to dyslipidemia and hypertension. The authors review the influence of polyphenols on endothelial dysfunction and atherosclerosis, but it would also be desirable to discuss about the effects of bioactive compounds on dyslipidemia and hypertension that represent the two major contributing risk factors for cardiovascular disease (CVD).
  • The paragraphs 6 and 7 need to be reorganized. The authors extensively discussed about HMGA1, which represent one of many possible mechanisms with which the bioactive compounds could exert their effects. I suggest to shorten the paragraphs regarding HMGA1 and also to report the data of other in vivo and in vitro studies that have investigated the molecular mechanisms with which bioactive compounds exert an effect on IR and related diseases.

Author Response

Authors: We thank the reviewer for the careful reading of the manuscript and his/her constructive remarks. We have accepted most of the comments to improve our manuscript. Please find below a detailed point-by-point response (reviewers' comments in black, our replies in blue italic).

REVIEWER 2

Major:

The role of polyphenols on glucose homeostasis is glossed over in this review. I suggest to insert a paragraph about the potential role of polyphenols in the prevention of type 2 diabetes mellitus. In the last few years several molecular and clinical studies have been conducted to evaluate the effects of flavonoids and their major food source on insulin resistance (IR). Please review this topic.

Authors: We have now covered the epidemiological and mechanistic evidence on the role of flavonoids in preventing type 2 diabetes in a dedicated section of the revised manuscript (Paragraph 3, “MedDiet flavonoids for preventing T2D”, lines 160-187).

IR is associated with increased risk for chronic kidney disease (CKD) in non-diabetic patients. Does the Mediterranean diet and its bioactive compounds have an effect on this IR-related disease?

Authors: We have now explored the bidirectional link between insulin resistance and CKD, as well as the protective role of the Mediterranean diet in people with kidney dysfunction, irrespective of the glycemic status, in a dedicate section of the manuscript (Paragraph 10, “MedDiet for managing CKD”, lines 624-665).

In the metabolic syndrome, IR is related to dyslipidemia and hypertension. The authors review the influence of polyphenols on endothelial dysfunction and atherosclerosis, but it would also be desirable to discuss about the effects of bioactive compounds on dyslipidemia and hypertension that represent the two major contributing risk factors for cardiovascular disease (CVD).

 Authors: We have now extensively discussed the role of the Mediterranean diet and its bioactive compounds on hypertension and plasma lipid abnormalities in two dedicated sections of the revised manuscript (Paragraphs 8 and 9, respectively, lines 539-623).

The paragraphs 6 and 7 need to be reorganized. The authors extensively discussed about HMGA1, which represent one of many possible mechanisms with which the bioactive compounds could exert their effects. I suggest to shorten the paragraphs regarding HMGA1 and also to report the data of other in vivo and in vitro studies that have investigated the molecular mechanisms with which bioactive compounds exert an effect on IR and related diseases.

Authors: We have now reassumed the clinical and preclinical evidence on the role of the Mediterranean diet and functional compounds against IR and IR-related diseases in two dedicated tables (pages 16-21). Besides HMGA1, other mechanistic findings have been discussed, including the activation of AMPK/sirtuin 1 by resveratrol, the modulation of PPARy, as well as other adipogenic factors, by anthocyanins and oleacein, and the reversion of a normal Akt/PKB signaling pathway by unsaturated fatty acids and hydroxytyrosol. However, we believe that targeting atherogenesis, tumorigenesis and glucose metabolism through a nutritional modulation of HMGA1, can raise new avenues for the investigators in this field, representing a strength point of our manuscript.

Round 2

Reviewer 2 Report

The authors replied adequately to my revisions. The review has been improved. 

This manuscript is a resubmission of an earlier submission. The following is a list of the peer review reports and author responses from that submission.

Round 1

Reviewer 1 Report

Mediterranean Diet Nutrients to Turn the Tide against Insulin Resistance and Related Diseases

The review summarizes the effectiveness and underlying mechanisms of major nutraceutical components present in Mediterranean diet against insulin resistance (IR)-related disorders, including type 2 diabetes mellitus (T2DM), non-alcoholic fatty liver disease (NAFLD), neurodegerative diseases and polycystic ovary syndrome (PCOS). The review is interesting and informative. Slight changes in terms of the sentence structure and contents are required to further improve the manuscript.

Minor comments:

In this review, the evidence for the health benefits of Mediterranean diet and its components are compelling, to the extent that it is almost too good to be true. Indeed, there are a lot of published data supporting these health benefits. However, it will still be good for the authors to include a section that discusses possible limitations, drawbacks or even flaws in the current research paradigm of Mediterranean diet. The authors may also want to point out important knowledge gaps which can serve as the future research focus to better understand the health effects of Mediterranean diet. Several recent clinical studies [1-4] which should are relevant to the review, are not covered. The findings of these human trials should be discussed. Page 2, line 58 - As a consequence, this eating pattern results is low in saturated fat … Certain sentences are very long and complex (i.e. Page 2 line 58-64; Page 7 line 287-291), making them hard to follow and understand. It is suggested that these sentences should be broken into shorter sentences so as to deliver the messages more effectively. Page 2, line 76 - should be T2D instead of TD2. Page 5, line 208 - As such, likewise like insulin sensitive peripheral tissues,  

References

Kaliora, A. C.; Gioxari, A.; Kalafati, I. P.; Diolintzi, A.; Kokkinos, A.; Dedoussis, G. V., The Effectiveness of Mediterranean Diet in Nonalcoholic Fatty Liver Disease Clinical Course: An Intervention Study. Journal of medicinal food 2019, 22, 729-740. Galan-Lopez, P.; Sanchez-Oliver, A. J.; Pihu, M.; Gisladottir, T.; Dominguez, R.; Ries, F., Association between Adherence to the Mediterranean Diet and Physical Fitness with Body Composition Parameters in 1717 European Adolescents: The AdolesHealth Study. Nutrients 2019, 12. de la Rubia Orti, J. E.; Garcia-Pardo, M. P.; Drehmer, E.; Sancho Cantus, D.; Julian Rochina, M.; Aguilar, M. A.; Hu Yang, I., Improvement of Main Cognitive Functions in Patients with Alzheimer's Disease after Treatment with Coconut Oil Enriched Mediterranean Diet: A Pilot Study. Journal of Alzheimer's disease : JAD 2018, 65, 577-587. Berti, V.; Walters, M.; Sterling, J.; Quinn, C. G.; Logue, M.; Andrews, R.; Matthews, D. C.; Osorio, R. S.; Pupi, A.; Vallabhajosula, S.; Isaacson, R. S.; de Leon, M. J.; Mosconi, L., Mediterranean diet and 3-year Alzheimer brain biomarker changes in middle-aged adults. Neurology 2018, 90, e1789-e1798.

Author Response

The review summarizes the effectiveness and underlying mechanisms of major nutraceutical components present in Mediterranean diet against insulin resistance (IR)-related disorders, including type 2 diabetes mellitus (T2DM), non-alcoholic fatty liver disease (NAFLD), neurodegerative diseases and polycystic ovary syndrome (PCOS). The review is interesting and informative. Slight changes in terms of the sentence structure and contents are required to further improve the manuscript.

Minor comments:

In this review, the evidence for the health benefits of Mediterranean diet and its components are compelling, to the extent that it is almost too good to be true. Indeed, there are a lot of published data supporting these health benefits. However, it will still be good for the authors to include a section that discusses possible limitations, drawbacks or even flaws in the current research paradigm of Mediterranean diet. The authors may also want to point out important knowledge gaps which can serve as the future research focus to better understand the health effects of Mediterranean diet.

Authors: As suggested by reviewer 1, we have now added a section named “Limitations and future research perspectives” (page 9, lines 355-381), addressing this point.

Several recent clinical studies [1-4] which should are relevant to the review, are not covered. The findings of these human trials should be discussed.

Authors: As suggested by reviewer 1, the indicated human trials are now discussed in the appropriate sections of the manuscript.

References

Kaliora, A. C.; Gioxari, A.; Kalafati, I. P.; Diolintzi, A.; Kokkinos, A.; Dedoussis, G. V., The Effectiveness of Mediterranean Diet in Nonalcoholic Fatty Liver Disease Clinical Course: An Intervention Study. Journal of medicinal food 2019, 22, 729-740.

Authors: See Lines 186-192 of the “Mediterranean diet and EVOO-derived secoiridoids for treating NAFLD” section.

Galan-Lopez, P.; Sanchez-Oliver, A. J.; Pihu, M.; Gisladottir, T.; Dominguez, R.; Ries, F., Association between Adherence to the Mediterranean Diet and Physical Fitness with Body Composition Parameters in 1717 European Adolescents: The AdolesHealth Study. Nutrients 2019, 12.

Authors: See Lines 64-74 of the “Introduction” section.

de la Rubia Orti, J. E.; Garcia-Pardo, M. P.; Drehmer, E.; Sancho Cantus, D.; Julian Rochina, M.; Aguilar, M. A.; Hu Yang, I., Improvement of Main Cognitive Functions in Patients with Alzheimer's Disease after Treatment with Coconut Oil Enriched Mediterranean Diet: A Pilot Study. Journal of Alzheimer's disease : JAD 2018, 65, 577-587.

Authors: See Lines 287-293 of the “Mediterranean diet and purple plant-derived anthocyanins extracts for neuroprotection” section.

Berti, V.; Walters, M.; Sterling, J.; Quinn, C. G.; Logue, M.; Andrews, R.; Matthews, D. C.; Osorio, R. S.; Pupi, A.; Vallabhajosula, S.; Isaacson, R. S.; de Leon, M. J.; Mosconi, L., Mediterranean diet and 3-year Alzheimer brain biomarker changes in middle-aged adults. Neurology 2018, 90, e1789-e1798.

Authors: See Lines 231-235 of the “Mediterranean diet and purple plant-derived anthocyanins extracts for neuroprotection” section.

Page 2, line 58 - As a consequence, this eating pattern results is low in saturated fat … Certain sentences are very long and complex (i.e. Page 2 line 58-64; Page 7 line 287-291), making them hard to follow and understand. It is suggested that these sentences should be broken into shorter sentences so as to deliver the messages more effectively. Page 2, line 76 - should be T2D instead of TD2. Page 5, line 208 - As such, likewise like insulin sensitive peripheral tissues, …

Authors: We have now checked for typos and grammar errors. Sentences highlighted in yellow (see lines 58-64 and 316-321 of the manuscript) have been shortened, in accordance to reviewers’ comments.

Reviewer 2 Report

The manuscript was prepared very well. No substantive errors found.

The selection of references is correct and I have no comments for it.

Figures were made precisely and have accurate descriptions.

Author Response

The manuscript was prepared very well. No substantive errors found.

The selection of references is correct and I have no comments for it.

Figures were made precisely and have accurate descriptions.

Authors: We thank reviewer 2 for the nice comments!

Reviewer 3 Report

There are to many references in the first paragraph of the introduction. In addition, some of them are very old and must be actualited.
There is no methodology section.
The article does not contribute anything new. There are already meta-analysis and reviews on the subject. And the conclusions are scarce and inaccurate.

Author Response

There are to many references in the first paragraph of the introduction. In addition, some of them are very old and must be actualited.

Authors: We have now updated and shortened the reference list in the “Introduction” section of the ms.

There is no methodology section.

Authors: As a narrative mini-review, and the way the manuscript is presented, the need for a “methodology section” is inappropriate.

The article does not contribute anything new. There are already meta-analysis and reviews on the subject. And the conclusions are scarce and inaccurate.

Authors: We are sorry for the reviewer's comment. However, unlike previous meta-analyses and reviews in this field, in our narrative mini-review we focused on the role that individual compounds, naturally present in Mediterranean foods, may have in the modulation of insulin resistance, a common feature of several pathological conditions linked to obesity. Our mini-review has mainly educational purposes. On the basis of the most recent evidence, our goal is to communicate to a wide audience of medical doctors, undergraduates and dieticians the therapeutic potential of the Mediterranean diet and synthetic plant-based polyphenol supplements, in the treatment of NAFLD, PCOS, type 2 diabetes and neurodegenerative diseases. Also, the potential molecular mechanism(s) involved in the modulation of insulin action by these molecules is discussed. Concerning the conclusions, a section named “Limitations and future research perspectives” (page 9, lines 355-381), has been added.